Development of a Taqman real-time PCR assay for rapid detection and quantification of Vibrio tapetis in extrapallial fluids of clams

Bidault Adeline adeline.bidault@univ-brest.fr
Richard Gaëlle G.
Le Bris Cédric
Paillard Christine
Laboratoire des Sciences de l’Environnement Marin (LEMAR), UMR 6539 UBO/CNRS/IRD/Ifremer, Université de Bretagne Occidentale , Plouzané , France
Thompson Cristiane
Electronic publication date: 2015 Dec 22
Publication date: 2015
Volume: 3
Electronic Location ID: e1484
Received 2015 Aug 21; Accepted 2015 Nov 18
Copyright: © 2015 Bidault et al.
Copyright year: 2015
Copyright holder: Bidault et al.
License: This is an open access article distributed under the terms of the Creative Commons Attribution License, which permits unrestricted use, distribution, reproduction and adaptation in any medium and for any purpose provided that it is properly attributed. For attribution, the original author(s), title, publication source (PeerJ) and either DOI or URL of the article must be cited.
License URL: https://creativecommons.org/licenses/by/4.0/

Keywords: Vibrio tapetis, virB4 gene, Taqman real-time PCR, Molecular diagnostic, Venerupis philippinarum, Marine pathogen, Brown ring disease

Funding: “Laboratoire d’Excellence” LabexMer ANR-10-LABX-19 French government Regional Council of Brittany University of Western Brittany This work was supported by the “Laboratoire d’Excellence” LabexMer (ANR-10-LABX-19) and co-funded by a grant from the French government under the program “Investissements d’Avenir,” and by a grant from the Regional Council of Brittany. This work was also funded by the University of Western Brittany. The funders had no role in study design, data collection and analysis, decision to publish, or preparation of the manuscript.

==============================
The Gram-negative bacterium Vibrio tapetis is known as the causative agent of Brown Ring Disease (BRD) in the Manila clam Venerupis (=Ruditapes) philippinarum. This bivalve is the second most important species produced in aquaculture and has a high commercial value. In spite of the development of several molecular methods, no survey has been yet achieved to rapidly quantify the bacterium in the clam. In this study, we developed a Taqman real-time PCR assay targeting virB4 gene for accurate and quantitative identification of V. tapetis strains pathogenic to clams. Sensitivity and reproducibility of the method were assessed using either filtered sea water or extrapallial fluids of clam injected with the CECT4600T V. tapetis strain. Quantification curves of V. tapetis strain seeded in filtered seawater (FSW) or extrapallial fluids (EF) samples were equivalent showing reliable qPCR efficacies. With this protocol, we were able to specifically detect V. tapetis strains down to 1.125 101 bacteria per mL of EF or FSW, taking into account the dilution factor used for appropriate template DNA preparation. This qPCR assay allowed us to monitor V. tapetis load both experimentally or naturally infected Manila clams. This technique will be particularly useful for monitoring the kinetics of massive infections by V. tapetis and for designing appropriate control measures for aquaculture purposes.

Introduction

The Manila clam Venerupis philippinarum (Adam & Reeve 1850) is the second most important aquaculture species in the world, after the oyster Crassostrea gigas (Gosling, 2008). V. philippinarum was introduced to French coasts in 1972 from the Pacific Coast of North America for economic purposes. The species progressively colonized the European coasts as it grows faster than the endemic European clam Venerupis decussatus (Flassch & Leborgne, 1994). In 1987, mass mortalities of cultured clams occurred in the main French production site located at Landéda (Brittany, France). Mortalities have been attributed to the Brown Ring Disease (BRD) due to the pathogenic Gram-negative bacterium Vibrio tapetis (Paillard & Maes, 1990).

V. tapetis first colonizes the periostracal lamina of the clam between the mantle and the shell. The proliferation of the pathogen in the periostracal lamina and extrapallial fluids (EF) inhibits the normal shell biomineralization process, resulting in a brown deposit of melanised matrix (conchiolin) on the inner surface of the valves (Paillard & Maes, 1994; Paillard & Maes, 1995a). The clam’s extrapallial fluids have been demonstrated to be a major compartment in the early stage of the defense process against the infection (Allam, Paillard & Maes, 1996; Allam, 1998). Indeed, extrapallial fluids contain numerous hemocytes (Allam, Paillard & Auffret, 2000) which are responsible for phagocytosis of micro-organisms (Allam & Paillard, 1998). Accumulation of hemocytes suggests an efficient defense system able to neutralize the pathogen before colonization of the extrapallial cavity and eventually tissues and could lead to septicemia (Allam & Ford, 2006). In this case, the penetration of V. tapetis into the hemolymph can provoke mortality before the clam exhibits BRD symptoms (Allam, Paillard & Ford, 2002; Paillard, Allam & Oubella, 2004). In a natural environment, the prevalence range of BRD in Manila clam has been estimated as 10–20% (Paillard et al., 2014).

Here, pathogenicity and virulence are defined as proposed in Casadevall & Pirofski (1999) and Sparling (1983), that is to say as the capacity of a microbe to cause damage to the, and the degree of pathogenicity respectively. A subtractive bank between two V. tapetis strains, i.e., the fish pathogen LP2 and the clam pathogen CECT4600T revealed that some genes are present only in the genome of the V. tapetis strain pathogenic to V. philippinarum (G Dias, 2015, unpublished data). Among these genes, virB4 existed only in the genome of the clam pathogen CECT4600T (accession number: KT382306). This finding suggests that some of these genes may be specifically associated with pathogenicity in the Manila clam. The genomes of several V. tapetis strains, including CECT4600T, were recently sequenced (G Dias, 2015, unpublished data), confirming the presence of one copy of the virB4 gene in the CECT4600T chromosome, using the MicroScope genomics platform (Vallenet et al., 2013). The virB4 gene encodes a protein involved in a large complex assigned to type IV secretion systems (T4SSs). In bacteria, secretion is essential for virulence and survival. Bacteria use T4SSs to translocate DNA and protein substrates across the cell envelope (Juhas, Crook & Hood, 2008; Low et al., 2014; Christie, Whitaker & González-Rivera, 2014), which contributes to genome plasticity and the evolution of pathogens through dissemination of antibiotic resistance and virulence genes (Fronzes, Christie & Waksman, 2009).

Classical methods currently used to identify Vibrio species associated with BRD were based on the cultivability of bacteria on non-selective synthetic medium and biochemical criteria (Borrego et al., 1996). Phylogenetic and molecular identification of Vibrionaceae, at the family and genus levels, can also be obtained by Multi Locus Sequence Typing using the rpoA, recA and pyrH genes (Thompson et al., 2005) or the gyrB gene (Le Roux et al., 2004; Yamamoto et al., 2000). Likewise, Single Specific Primer-PCR (SSP-PCR) amplification related to 16S rRNA identity (Paillard et al., 2006) or molecular typing and fingerprinting methods (Romalde et al., 2002; Rodríguez et al., 2006) have been developed. Drummond and collaborators (2006) showed that the combination of shell valve analysis with the SSP-PCR assay of Paillard et al. (2006) proved to be most sensitive but was time-consuming and not specific enough and sensitive enough to detect V. tapetis pathogens in clams. These methods are therefore inadequate for monitoring individual clams and above all, inappropriate to detect asymptomatic infected clams.

The aim of this study was to develop a rapid and accurate detection method for Vibrio tapetis, the causative agent of BRD, which is of growing interest due to the increased prevalence of the disease and the high commercial value of clams. Taqman real-time PCR protocols have previously been carried out for V. aestuarianus (Saulnier, De Decker & Haffner, 2009; McCleary & Henshilwood, 2015) and V. harveyi (Schikorski et al., 2013). In this paper, we developed a Taqman real-time PCR assay for specific and rapid V. tapetis detection and quantification from extrapallial fluids of clams, and validated it on both pure cultures and through an experimental infection of V. philippinarum by a virulent V. tapetis strain.

Materials and Methods

Bacterial strains and culture conditions

Bacteria belonging to twelve Vibrio species and seventeen strains of V. tapetis isolated from clams and fishes were used in this study (Table 1) as positive and negative controls to check species specificity of the designed primers. Some reference strains were purchased from CIP and LMG collections. V. anguillarum strain 775 was supplied by JH Crosa (Frans et al., 2011; Crosa, Schiewe & Falkow, 1977). The Norwegian S2-2 strain was kindly offered by S Mortensen (2009, unpublished data) from the Institute of Marine Research at Bergen in Norway. Finally, the FPC 1121 strain was graciously provided by T Matsuyama (Matsuyama et al., 2010). The other strains were available from the LEMAR collection.

Table 1 Bacterial strains used in this study isolated from different hosts and origins, and specificity of Taqman qPCR method for the detection of virB4 gene in strains pathogenic for clams.

Strain	Vibrio species	Source and location isolation	Strain reference	Virulencea	qPCRb	
CECT4600T	V. tapetis	Venerupis philippinarum, Landéda, France	Paillard & Maes, 1995b	+d	+	
FPC 1121	V. tapetis	Venerupis philippinarum, Japan	Matsuyama et al., 2010	+	–	
IS 1	V. tapetis	Venerupis philippinarum, Landéda, France	Borrego et al., 1996	+d	+	
IS 5	V. tapetis	Venerupis philippinarum, Landéda, France	Borrego et al., 1996	+d	+	
IS 7	V. tapetis	Venerupis philippinarum, Quiberon, France	Borrego et al., 1996	+d	+	
IS 8	V. tapetis	Venerupis aurea, Quiberon, France	Borrego et al., 1996	+d	+	
IS 9	V. tapetis	Cerastoderma edule, Quiberon, France	Borrego et al., 1996	+d	+	
P16B	V. tapetis	Venerupis philippinarum, Morbihan Gulf, France	Allam, Paillard & Ford, 2002	+d	+	
RD 0705	V. tapetis	Venerupis decussatus, Galicia, Spain	Novoa et al., 1998	+d	+	
RP 11.2	V. tapetis	Venerupis philippinarum, Landéda, France	Borrego et al., 1996	+d	+	
RP 2.3	V. tapetis	Venerupis philippinarum, Landéda, France	Borrego et al., 1996	+d	+	
RP 8.17	V. tapetis	Venerupis philippinarum, Landéda, France	Borrego et al., 1996	+d	+	
RP 9.7	V. tapetis	Venerupis philippinarum, Landéda, France	Borrego et al., 1996	+d	+	
UK6	V. tapetis	Venerupis philippinarum, Poole Harbour, UK	Allam, Paillard & Auffret, 2000	+d	+	
S2-2	V. tapetis	Solea Solea, Norway	S Mortensen, 2009, unpublished data	−c	–	
LP2	V. tapetis	Symphodus melops, Bergen, Norway	Jensen et al., 2003	−d	–	
HH6087	V. tapetis	Hippoglossus hippoglossus, Inverailort, UK	Reid et al., 2003	−d	–	
LMG 20012T	V. tasmaniensis	Salmo Salar L., Tasmania Australia	Thompson, Thompson & Swings, 2002	−d	–	
LMG 4042T	V. splendidus	marine fish	Le Roux et al., 2004	−d	–	
LMG 19703T	V. shilonii	Oculina patagonica, Mediterranean sea	Kushmaro et al., 2001	nd	–	
LMG 20539T	V. kanaloe	Ostrea edulis larvae, France	Thompson, 2003	nd	–	
CIP 107166T	V. lentus	cultivated oyster, Spain	Macián et al., 2001	nd	–	
LMG 16745T	V. chagasii	marine fish	Le Roux et al., 2004	nd	–	
775 (ATCC 68554)	V. anguillarum	Oncorhynchus kisutch, US Pacific Coast	Crosa, Schiewe & Falkow, 1977	nd	–	
02/041	V. aestuarianus	Crassostrea gigas, Argenton, France	Garnier et al., 2007	nd	–	
P9	Halomonas sp. 33	Venerupis philippinarum, Marennes, France	Paillard et al., 2006	−d	–	
CF6	V. splendidus	Crepidula formicata	Choquet, 2004	−d	–	
GM4	Vibrio. sp.	Venerupis philippinarum, Morbihan Gulf, France	Paillard et al., 2006	−d	–	
ORM4	V. harveyi	moribund abalone, France	Austin & Zhang, 2006	−c	–	
Notes.

a Virulence in vivo on Venerupis philippinarum.

b Real-time PCR results.

c Unpublished.

d Published on Choquet thesis Choquet, 2004.

nd Not determined.

The V. tapetis CECT4600T strain was used for the experimental infection as the reference strain (Paillard & Maes, 1995b; Borrego et al., 1996).

All bacterial strains were cultured in Zobell medium (pastone 4 g/L, yeast extract 1 g/L, sea salt 30 g/L at pH 7.4) enriched with iron phosphate (0.1 g/L) at 18 °C during 18 h under constant shaking at 180 rpm (Infors HT®) (Balboa et al., 2012).

Clams, experimental infection and extrapallial fluids collection (EF)

Two year old Venerupis philippinarum were provided by Fabien Fonteneau in Marennes-Oleron (“Les Claires de Bonsonge”®, EARL, brood stock producer). A first health diagnostic was performed in situ on 50 clams to ensure absence of BRD, prior to the sampling effort for the bacterial challenge. They were transferred to the Ifremer’s facilities (Laboratoire de Physiologie des Invertébrés, LEMAR, Plouzané) for six days quarantine with chloramphenicol (8 mg/L; Sigma Aldrich, St. Louis, MO, USA) at 13 °C.

For the duration of the experiment, clams were split randomly into eighteen 100 L-tanks with air-lift systems and a complete water-renewal every two days. During the whole experiment, clams were daily fed an algal ration (maintenance ratio from FAO, 2004) of two algae commonly used in aquaculture (Isochrysis affinis galbana and Chaetoceros calcitrans) (Figs. 1 and S1A).

Figure 1 Schematic view of the infection procedure.

Four weeks after algal-conditioning, clams were exposed to air for 12 h in the experimental room and replaced into water just before injection to facilitate valve opening. Seventy two clams were injected, in the extrapallial cavity, with 100 µL of Filtered Sea Water (FSW), 72 were injected with 100 µL of V. tapetis CECT4600T fresh suspension, at a 107 cells per mL density and the last 72 clams were not injected (Fig. 1).

Twelve samples from each three different conditions (a total of 36 clams) were sampled at six different time points (216 clams sampled in total): 0d, 1d, 2d, 7d, 14d and 30d (respectively 0—not injected, 1, 2, 7, 14 and 30 days post injection). For each clam, 500 µL of extrapallial fluids were collected close to the shell under mantle, using a syringe fitted with a 25-G needle. The fluids were immediately flash frozen in liquid nitrogen and stored at −80 °C until DNA extraction.

Extrapallial fluids from an additional 40 clams were withdrawn and pooled to constitute the bacteria dilution range. Fifteen mL of the resulting pool were filtered from 80 µm to 0.22 µm, with intermediary filtrations at 10 µm, 1 µm and 0.45 µm, to obtain extrapallial fluids free of bacteria.

The presence of V. tapetis CECT4600T was determined and quantified in collected samples by real-time PCR using the appropriate standard curve.

BRD diagnostic method

After fluid collection, clams’ shells were retrieved at each sampling in order to diagnose BRD occurrence. Images of inner shells were obtained using a 50 mm Canon® macroscopic lens and analyzed using image analysis software (Visilog® 6.6) (Richard et al., 2015, unpublished data). Subsequently, clams exhibiting brown ring deposit surfaces were defined as BRD positive (BRD+) in this study (Fig. 2). Those that were visibly healthy were reported as BRD negative (BRD−).

Figure 2 Photography of (A) BRD- clam and (B) BRD+ clam.

From Richard et al., 2015, unpublished data.

Total DNA extraction from bacterial culture and from EF

DNA extraction was performed using the QIAamp DNA mini kit (Qiagen) for both bacterial cultures and total extrapallial fluids. 450 µL were centrifuged at 10,000 g at 4 °C for 10 min. Pellets were deproteinized in a hot dry bath at 56 °C by addition of 180 µL of ATL Buffer supplemented by 20 µL of proteinase K during one hour. DNA extractions were then performed according to supplier instructions. Finally, DNA were eluted in 200 µL of ultra-pure water and stored at −20 °C until use. DNA yield and purity were determined by spectrophotometry (Quantifluor dsDNA kit; Promega, Madison, WI, USA; and POLARstar Omega microplate spectrophotometer; BMG Labtech, Orgenberg, Germany).

Enumeration of V. tapetis by spectrophotometry

In order to accurately inoculate CECT4600T at the target dilution of 107 cells/mL during the experiment, enumeration of bacteria was performed to measure the absorbance at 492 nm on a Multiskan spectrophotometer (Fisher Scientific, Hampton, NH, USA). Vibrio tapetis culture density was calculated according to the formula 1.3 109 × DO − 3.6 107 CFU/mL (Choquet, 2004; Le Bris et al., 2015).

For the dilution ranges of bacteria (described below), an early stationary phase culture of CECT4600T was enumerated, and was first diluted in Filtered SeaWater (FSW) to obtain a 108 cells/mL suspension. Bacteria were then serially diluted in FSW in a final volume of 500 µL from 2.25 107 to 0.565 101 cells/mL (ten-fold dilutions until 2.25 101 and half dilutions for the last two) to generate the standard curve of bacteria. DNA was extracted from 450 µL of each diluted suspension. For the second standard curve in extrapallial fluids, FSW was replaced by filtered EF prepared previously. DNA extractions were also performed on 450 µL of sterile EF in the same conditions in order to establish a negative control for the standard curve.

PCR primer and fluorogenic probe design

Oligonucleotide sequences used in this study are listed in Table 2. The virB4 primers and probe were developed to target exclusively the virB4 gene existing in the pathogenic Vibrio tapetis CECT4600T (Genbank accession number: KT382306). Using Primer-3 software, several primers pairs were designed and tested with DNA samples of different species and strains of the Vibrio genus available as positive and negative controls on a 9700 ABI® thermocycler (Applied Biosystems, Foster City, CA, USA) (Table 1). The choice for the best primer pair was determined, with optimal concentration and reaction conditions for PCR amplification compatible with the hydrolysis Taqman probe (Table 2). The virB4 probe was dually labeled with 5′-reporter dye 6-FAM (wavelength emission at 502 nm) and a downstream 3′-quencher dye TAMRA.

Table 2 Nucleotide sequences and melting temperatures (Tm) of primers and probe designed for real-time PCR reaction, targeting the virB4 gene.

Description	Name	Sequence (5′–3′)	Tm (°C)	
Forward primer	virB4-F3	TTA-AAA-GTG-GCG-GAG-GAA-TG	58	
Reverse primer	virB4-R3	AAG-CTC-TGC-ATC-GGT-TAG-GA	60	
Taqman probe	virB4-P1	CGA-GTA-CCA-ACA-TGC-CTT-CCC-GT	53.7	

The selectivity of the primers was checked using the BLAST algorithm of the NCBI database (http://www.ncbi.nlm.nih.gov/), assuring their specificity for the virB4 region, without homology to other known sequences described in GenBank and EMBL databases. Oligonucleotides were also aligned to the CECT4600T and LP2 genomes to ensure that they selectively amplified only the virB4 gene. Primers and probe were purchased from Eurogentec (Angers, France). The expected length of the amplicon deduced from nucleotide sequence was 173 bp, and the selectivity was also theoretically assessed by BLAST.

Quantitative real-time PCR (qPCR)

Real-time PCR was performed on a LightCycler 480 Instrument (Roche Diagnostics, Mannheim, Germany) using LightCycler 480 Probe Master Mix based on Taqman detection (Roche Diagnostics). Each real-time PCR experiment included technical triplicates, in a final volume of 15 µL. Each reaction contained 5 µL of DNA template, 0.5 µM of each primer, 0.1 µM of hydrolysis probe virB4-P1 and 7.5 µL of LC480 Probe Master Mix 2X. A single initial denaturation step of 10 min at 95 °C was followed by 45 cycles of 95 °C for 10 s (denaturation), 54 °C for 20 s (annealing) and 72 °C for 1 s to measure the fluorescence signal. Finally, a cooling step at 40 °C during 10 s was included. The results were analyzed with Roche LightCycler 480 software. Threshold cycle (Ct) value corresponds to the PCR cycle number at which an increase in reporter fluorescence above a baseline signal was first detected, after background subtraction. Negative controls with molecular biology grade water as template were performed in each run. For quantitation, dilution ranges of V. tapetis were tested in triplicates across multiple orders of magnitude described above. Results were analyzed by linear regression to calculate the slope. The PCR amplification efficiency (E) was measured according to E = [101/(−slope)] − 1, using the Roche LightCycler 480 software. Standard curve data points were used to determine real-time PCR quantification and detection limits through the assessment of the variance (standard deviation), measured at each dilution standard. In conformity with ISO 16140, the limit of detection (LOD) was calculated in accordance with 95% of 20 tested replicates giving positive amplification results (Bustin et al., 2009). The repeatability (intra-assay variance) was estimated using triplicates of each template to assess the precision of the method. The reproducibility (inter-assay variance) was calculated using standard deviation (SD) of two virB4 gene standard concentrations used for each run to assess the variation between runs.

Results

Selectivity of the real-time PCR protocol

Sequence alignments of the real-time Taqman PCR amplicon showed no cross reactivity with others species when compared using the BLAST analysis program. Only one theoretical partial cross reactivity with V. tasmaniensis (KP795691.1, 78% of identity with differences in 3′ end terminals of both forward primer and Taqman hydrolysis probe) appeared with BLAST analysis when comparisons were extended (more dissimilar sequence search settings). The species selectivity of the real-time PCR assay developed for the identification of V. tapetis was empirically evaluated in this study on 17 V. tapetis strains (Table 1), including two reference strains CECT4600T and LP2, isolated from the Manila clam Venerupis philippinarum and the fish Symphodus melops,  respectively, and 12 marine pathogens, mostly Vibrio, except for one (Halomonas sp.). Real-time PCR results obtained showed that all virulent V. tapetis strains tested, i.e., able to reproduce BRD after injection into the pallial cavity (Choquet et al., 2003; Paillard, Allam & Oubella, 2004), were positive (except for FPC1121 strain, a Japanese V. tapetis), whatever the geographical origin of the bacterial strain. For example, UK6 and IS9 strains were clearly positives though they were isolated from V. philippinarum in United Kingdom and from C. edule in France, respectively. Moreover, the LP2 reference strain which is a pathogen for S. melops (Jensen et al., 2003) but not for the Manila clam (Paillard, Allam & Oubella, 2004), did not show any signal for the virB4 gene (Fig. 3). No amplification was detected with bacterial strains belonging to the other Vibrio species (n = 12), even for the closest phylogenetic neighbors to V. tapetis, among tested strains (V. tasmaniensis, V. spendidus, V. lentus) (Thompson, Thompson & Swings, 2002; Le Roux et al., 2004; Sawabe et al., 2013; Al-saari et al., 2015), in agreement with the empirical and theoretical results generated by BLAST.

Figure 3 Visualization of the PCR product in agarose gel obtained with qPCR virB4 assay for representative strains of Vibrio, i.e., which were tested positive and negative for BRD development after an infection experiment.

Lanes MT corresponds to the BenchTop DNA ladder (Promega, Madison, WI, USA). T-H2O represents the water negative control.

Positive signals obtained by real-time PCR experiments for bacterial strains were checked by electrophoresis. The amplicon observed at 173 bp, corresponds to the expected size, calculated from the nucleotide sequence of the virB4 gene (Fig. 3).

Sensitivity of the real-time PCR assay

Standard curves for virB4 gene quantification were generated in parallel in filtered sterile water (FSW) and total extrapallial fluids (EF) with pure bacterial suspension of V. tapetis CECT4600T strain of known concentrations, determined by spectrophotometry and checked by the Malassez counting method. Ten-fold bacterial dilutions ranging from 2.25 107 to 2.25 102 cells mL−1, and two last half dilutions to 0.565 101 cells mL−1 were prepared in these two diluents. Since only one copy of the chromosomal virB4 gene is present per bacteria, the threshold cycle (Ct) values deduced from real-time PCR amplifications on purified DNA extracts were plotted to the number of bacteria initially present in PCR templates.

Because similar slopes were achieved for quantification curves of each diluent in this study (data not shown), the standard curve in EF was chosen for analyzing the experimental infection samples as this diluent corresponded to the in situ conditions of the samples, namely V. tapetis in EF.

The standard curve reliably showed linearity across 8 orders of magnitude, from 2.25 107 to 1.125 101 cells mL−1. The final dilution (0.565 101 cells mL−1) produced less consistent results, and was not retained for the calculation of the standard curve. After extractions, genomic DNA was checked by spectrophotometry and corresponded to values ranging from 900 pg µL−1 to 156 pg µL−1. According to the Ct values obtained in triplicates, quantification curve exhibited an excellent linear regression with an r2 correlation coefficient of 0.99 and a PCR efficacy of 103% (Fig. 4) calculated in the linear zone according to the MIQE guidelines (Bustin et al., 2009). The intra-assay variance was evaluated from 0.01 and 0.32 from all the samples tested in triplicate. The reproducibility, calculated using the standard deviation of Ct values generated from two standard concentrations of 2.25 102 cells mL−1 and 2.25 106 cells mL−1 from different runs, corresponded to 0.14 and 0.10 respectively.

Figure 4 Standard curve for the detection and quantification of the virB4 gene by Taqman real-time PCR, in dilution range of EF samples artificially spiked with CECT4600T bacterial strain.

Standard curve was generated by plotting the log cell number of bacteria present in PCR DNA template against Ct values.

The threshold sensitivity of this method in targeting the presence of virB4 gene and quantification is given by the lower bacterial concentration detected in the linear zone (at least 95% of 20 tested replicates), and corresponds to the limit of detection (LOD) of V. tapetis of 1.125 101 bacteria mL−1.

Specificity of the method: kinetics of infection during Manila clams challenge

The quantification of CECT4600T DNA in extrapallial fluids sampled from infected clams was estimated by reporting Ct values to the standard quantification curve of the virB4 gene previously established for EF. The limit of detection established previously corresponded to a cell density of 1.125 101 bacteria per mL of extrapallial fluids.

Overall during the experiment, each sampling time of CECT4600T injected clams exhibited at least two positive individuals by the real-time PCR assay (Fig. 5). Only 24 h after injection, nine individuals among twelve were spotted with the virB4 detection protocol. At the same time, all of them were BRD−. It was noticeable that at 7 days post-injection only two real-time PCR positive animals were detected, but with a high V. tapetis load, at nearly 107 cells per mL in EF (6.53 106 cells mL−1). As regards the BRD among CECT4600T-injected clams identified as positive by real-time PCR, one individual at 2 days and one at 7 days were classified as BRD+. At 14 and 30 days post-injection, four clams at each time were positive according to the qPCR assay and they were all BRD+. All these V. tapetis-positive individuals showed a bacterial load above 102 cells per mL in extrapallial fluids, with a maximum of 1.11 104 cells per mL in extrapallial fluids.

Figure 5 Kinetics of clam infection by CECT4600T V. tapetis strain by virB4 real-time PCR in extrapallial fluids sampled at 0, 1, 2, 7, 14 and 30 days post-injection.

0d means not injected. * corresponds to BRD+ clam.

The virB4 gene was also detected in several non-injected individuals: three animals on the day of infection; and one each at 2, 7 and 30 days during the experiment (Fig. 6 and the points 0d on Figs. 5 and 7). The animals were BRD− at 0 and 30 days of the experiment whereas the two positive virB4 clams (2d and 7d on Fig. 6) displayed clinical signs of BRD.

Figure 6 Kinetics of non-injected clams by virB4 real-time PCR in extrapallial fluids sampled at 0, 1, 2, 7, 14 and 30 days of sampling during the experiment.

* corresponds to BRD+ clam.

Figure 7 Kinetics of FSW-injected clams by virB4 real-time PCR in extrapallial fluids sampled at 0, 1, 2, 7, 14 and 30 days of sampling during the experiment. 0d means not injected.

* corresponds to BRD+ clam.

Regarding the detection of virB4 gene by real-time PCR assay on animals injected with FSW (Fig. 7), one individual tested positive on the day of the infection. Six animals were identified as positive 24 h after the injection of FSW. Two days post-injection, two clams among 12 were positive according to the Taqman protocol and at later times, only one clam was tested positive at 7, 14 and 30 days post-injection.

Discussion

In this study, we developed a rapid and accurate real-time PCR process for the detection and quantification of V. tapetis pathogens for clams in extrapallial fluids of the Manila clam. This protocol was designed using a pair of primers and a Taqman probe targeting the virB4 gene of V. tapetis, encoding a protein engaged in a large complex of type IV secretion systems.

Several publications dealing with rapid and specific molecular identification of Vibrio species use PCR techniques. Various PCR assays have been published (Thompson et al., 2005; Paillard et al., 2006; Rodríguez et al., 2006) that have investigated Vibrio tapetis. These methods, based on 16S rRNA sequences are less reliable and can be time-consuming when associated with the BRD diagnostic necessary to achieve Vibrio tapetis identification (Drummond et al., 2006). Moreover, these protocols do not allow quantification of the bacteria.

According to our knowledge, most real-time PCR assays targeting detection and quantification of Vibrio species have been developed in Vibrio spp. impacting public health risk, e.g., Vibrio cholerae (Gubala, 2006; Mehrabadi et al., 2012), Vibrio vulnificus (Garrido-Maestu et al., 2014). Only rarely have studies dealt with marine pathogenic Vibrio, e.g., V. aestuarianus and V. harveyi (Saulnier, De Decker & Haffner, 2009; Schikorski et al., 2013). In this work, we have performed for the first time a real-time PCR protocol to detect V. tapetis bacteria.

The specificity of primers and Taqman probe have been demonstrated here with the real-time PCR assays carried out on DNA samples extracted from pure cultures of various bacterial strains belonging to the V. tapetis group. No signal was obtained with other species close related to V. tapetis (belonging to V. splendidus or V. tasmaniensis), even at high threshold cycle values (Ct > 45). Positive fluorescence signals were acquired with all V. tapetis strains virulent for V. philippinarum, except the FPC1121 Japanese strain (Table 1). We suggest that this strain lacks the virB4 gene and we could speculate that this Japanese strain uses a different secretion system from the type IV to translocate substrate for its virulence (G Dias, 2015, unpublished data).

The standard curves generated by real-time PCR in filtered sea water and extrapallial fluids demonstrated excellent coefficients of correlation for the primers and Taqman probe used. Furthermore, both standard curves were stackable indicating that the DNA extraction procedure for extrapallial fluids samples was satisfactory. Taking into account the dilution factor employed in this study to extract DNA from biological samples, we established the calculated threshold sensitivity of the method at 2.8 10−1 cells per well. Thus, the limit of detection of the virB4 real-time PCR assay corresponds to 11.25 bacteria per mL of FSW or EF of clam. This limit of detection is very low compared to V. harveyi and V. aestuarianus detections methods, which are at 18 and 1.6 bacteria per well respectively (Schikorski et al., 2013; Saulnier, De Decker & Haffner, 2009) and suggests our technique is useful for detecting both a weak and massive infections of V. tapetis in Manila clam.

We used the real-time PCR developed in this study to monitor the kinetics of V. tapetis experimental infection in clams. During bacterial challenge, early stages of the infection showed a high number of individuals infected and high bacteria load (until 7 days post-injection), confirming that extrapallial fluids are involved in the first steps of infection (Allam, Paillard & Maes, 1996), where hemocytes contribute to defense against V. tapetis by phagocytosis (Allam & Paillard, 1998). At 7 days post-injection, only two individuals were detected as positive by the real-time PCR assay, suggesting that animals had efficiently defended against the pathogen. This hypothesis is corroborated by Richard and collaborators (2015, unpublished data). This result also agrees with Paillard & Maes (1994), who showed that the first BRD symptoms appeared in almost all the clams seven days post-injection, bearing out the reaction of hosts facing a pathogen injection. Indeed, the conchiolin matrix was developed to trap encompassing bacteria into the inner surface of the shell, inducing a decrease of circulating bacteria in the EF, and thus a lower concentration detected by the Taqman assay. Then, 14 and 30 days after injection, CECT4600T was again found in extrapallial fluids but the load was less than 7 days post-injection. Paillard and collaborators (2014) argue that an increase of V. tapetis in the host’s fluids in advanced stages of the disease is due to the weakening of the clam, despite food intake and shell repair.

Moreover, regarding individuals detected as positive among animals injected with FSW, we observed that they were mainly detected 1 day after injection. These results imply an effect of injection on animals which were supposedly infected by V. tapetis, present in their environment before sampling, despite the quarantine stage. It has already been shown that injection or handling could stimulate bacterial proliferation (Le Bris et al., 2015; Jean et al., in press). Focusing on non-injected clams, i.e., at 0d for V. tapetis- and FSW-injected clams and all the sampling times for non-injected clams, the interpretation that V. tapetis was present in individuals before sampling is corroborated: 3 individuals among 36 were already carrying bacteria on the day of injection. During the remaining time, 3 non-injected animals were detected as positive by real-time PCR protocol, while 11 animals displayed clinical signs of BRD among 72 sampled (15%), suggesting that the host has already mounted a defense response against pathogens before the experimental challenge. This corresponds to the natural prevalence of the disease described in literature (Paillard et al., 2014).

To conclude, we developed in this study a rapid, specific and individual method to detect and quantify V. tapetis. This protocol is based on a real-time PCR assay and is suitable for field and hatchery animals. Indeed, this virB4 real-time PCR assay is easy to implement because it does not require crushing and grinding and allows individual assays considering inter-individual variability. This protocol allows for early detection of the disease, especially to assess visibly healthy clams (BRD−), bearing in mind the threshold of detection, and will be very useful in helping prevent massive infection in clams, notably in clam aquaculture.

Supplemental Information

Figure S1 Photography of the 18 incubation tanks

Click here for additional data file.

We would like to thank the company SATMAR for providing us with clams during the first steps of optimization of the method, and “Les Claires de Bonsonge” for supplying us clams for the experimental challenge. We are thankful to Valérie Barbe and Claudine Médigue from LABGeM and the National Infrastructure “France Genomique” for sequencing CECT4600T strain and Annick Jacq for expert annotation. We thank Didier Mazel and Frédérique Leroux, who were in charge of the Vibrioscope project within MaGe. We also thank Prof. Vianney Pichereau for his critical reading and advice on the manuscript, and Ewan Harney, a native speaker, for English corrections.

Additional Information and Declarations

Competing Interests

Author Contributions

Patent Disclosures

DNA Deposition

Data Availability

The authors declare there are no competing interests.

Adeline Bidault conceived and designed the experiments, performed the experiments, analyzed the data, wrote the paper, prepared figures and/or tables.

Gaëlle G. Richard performed the experiments, analyzed the data, prepared figures and/or tables, reviewed drafts of the paper.

Cédric Le Bris performed the experiments.

Christine Paillard contributed reagents/materials/analysis tools, reviewed drafts of the paper.

The following patent dependencies were disclosed by the authors:

V. tasmaniensis genbank accession number KP795691.1.

The following information was supplied regarding the deposition of DNA sequences:

GenBank accession number: KT382306.

The following information was supplied regarding data availability:

This work did not generate any raw data.

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
