# Peer review of "Development of a Taqman real-time PCR assay for rapid detection and quantification of Vibrio tapetis in extrapallial fluids of clams"

_PeerJ, doi:10.7717/peerj.1484_

## Round 0.1 · original submission · Major Revisions

Dear Authors,
I agree with the reviewers' comments. In particular the suggestions related to methods (standard curve, all parameters mentioned by review 1 and 2, clams experimental infection and extrapallial fluids collection (EF), descriotion of BRD diagnostic method, PCR primer and fluorogenic probe design, BRD prevalence... have to be clearly defined in methods) and a clear improvement of the discussion and conclusion should be adressed. Please, make sure you address all the Editor and reviewers' suggestions or requirements, point-by-point.

Reviewer 1 ·

Basic reporting

No Comments

Experimental design

This work aims to contribute at managing the problem of V. tapetis derived BRD in clams. It proposes a detection and quantification tool based in Taqman RT-PCR, which is suited to meet the author’s target. Is this sense it is welcome to the field. But to fully achieve its goals the work is expected to conform to the International Organization for Standardization ISO 16140 validation recommendations. My main concern is with the standard curve. It was not built employing PCR copies from purified plasmid DNA. The main parameters to be evaluated in this approach were absent or could not be recognized: selectivity, linearity, efficiency and Limit of Detection (LOD). Such parameters have to be clearly defined in methods so that the corresponding results can be reported and recognized. Besides ISO 16140, a good reference is Bustin et al., 2009 (The MIQE Guidelines: Minimun information for publication of quantitative Real-Time PCR experiments).
More comments in Validity of the Findings and General.

Validity of the findings

As previously mentioned, the standard curve should be built with PCR copies from purified plasmid DNA. The standard curve is expected to be presented as Ct x PCR plasmid copy number. Once you build a Ct x PCR plasmid copy number curve, the following parameters are expected to be analysed.
Efficiency: The interpretation of the results by linear regression for slope analysis and the PCR amplification efficiency according to E= [101/(-slope)] -1 are expected. PCR efficacy (103%?) was reported in results but not defined in methods.
Linearity: it is expected that you report for how many orders of magnitude the standard curve reliably shows linearity. Please mention the parameter.
Limit of Detection (LOD): according to ISO 16140 it is required that 95% of 20 tested replicates return positive amplification results. This fulfillment has to be clear in text (say you tested in methods and tell the result in results).
Selectivity (a method is selective if it can be used to detect the target and that a guarantee can be provided that the detected signal can only be a product by the specific target): you did test the selectivity by BLAST analysis and empirically. Let clear to the readers by naming this parameter in methods and reporting in results. You employed specificity. The specificity is the degree to which the reaction is affected by the other components present in a multi-component sample. In other words, the ability of a method to measure a given analyte within the sample without interference from non-target components or background noise.
Other suggested literature: McCleary & Henshilwood, 2015 (Novel quantitative TaqMan® MGB real-time PCR for sensitive detection of Vibrio aestuarianus in Crassostrea gigas); Molecular Diagnostics: current research and applications, 2014 (Eds. Jim Huggett and Justin O'Grady. Caister Academic Press).

Additional comments

Despite the need for ISO 16140 conformity (technical and nomenclature) the work is certainly useful and will be welcome in both the academia and industry realms.
Minor modifications are suggested in text.
Line 36: biomineralization
Line 37: The clam’s extrapallial …
Line 43: pallial
Line 45: …the prevalence range of BRD in Manila clam has been estimated as 10 to 20%.
Line 49: Among these genes, virB4 was …
Line 55: … protein substrates across the cell envelope 16 17 18 , which contributes to…
Line 62: … Single Specific Primer-PCR (SSP-PCR)…
Lines 64-5: … proved to be the most sensitive but was time-consuming and not enough specific and sensitive to detect V. tapetis…
Along text correct Tm: Tm (m is subscript and italicized)
Lines 195-8: The species specificity of the real-time PCR assay developed for the identification of V. tapetis was evaluated in this study on sixteen V. tapetis strains and 15 marine pathogens, mostly vibrios, except for one…
Line 200: …were positive (except for FPC1121 strain, a Japanese V. tapetis),…
Line 204- : No amplification was detected with bacterial strains belonging to the other Vibrio species (n=12), even for the closest phylogenetic neighbors to V. tapetis , among tested strains (V. tasmaniensis, V. splendidus and V. lentus) 39 22 Sawabe et al., 2013 (Updating the Vibrio Clades Defined by Multilocus Sequence Phylogeny: Proposal of Eight New Clades, and the Description of Vibrio tritonius sp.nov; Al-saari et al., 2015 (Advanced Microbial Taxonomy Combined with Genome-Based-Approaches Reveals that Vibrio astriarenae sp. nov., an Agarolytic Marine Bacterium, Forms a New Clade in Vibrionaceae)
Line 283: EMSF (???)
Line 330- : … the virB4 gene and we could speculate that the Japanese strain uses a different secretion system than the type IV to translocate substrate for his virulence.

Methods:
Clams, experimental infection and extrapallial fluids collection (EF)
Describe BRD diagnostic method
PCR primer and fluorogenic probe design
Please inform the accession number(s) for VirB4 target gene.
BRD prevalence
Please provide some images of symptomatic BRD and BRD negative (one of each at least). If you wish to include a photo of the incubation tank and a photo of the infection procedure (recommended) you can provide all photos as supplementary figures.

Table 1: This table misses strain RP 8.17

Figure 1 (legend)
Visualization of the PCR product in agarose gel obtained with qPCR virB4 assay for representative strains of Vibrio, ie which tested positive and negative for BRD development after the infection challenge. Lanes MT corresponds to the BenchTop® DNA ladder (Promega®).

Reviewer 2 ·

Basic reporting

The study is very interesting well written and the finds are significant. However, some important points should be reevaluated by the authors.

You mentioned at line 307 “Then, 14 and 30 days after injection, CECT4600T was again largely found in extrapallial fluids but the load was less than 7 days post-injection”. It seems you are super estimated this analysis because at 7 days infection you have only two sample detected and only one has very high vibrio cells detected.
You mentioned “Seven days after injection, BRD symptoms appeared in almost all the clams, bearing out the reaction of host faced to the pathogen injection” but vibrio detection was reported only in 4 of 12 samples.
I believe the authors should better explore the discussion of the data. For instance, why it is possible to detect vibrio strain only in some symptomatic clam after 7 days?
No discussion is mentioned about the low capability of vibrio detection in non-injected clams (Figure 4) even in symptomatic samples (30 d) or before that.
You have concluded that : This protocol allows for early detection of the disease, especially to assess asymptomatic clams (BRD negative) but for instance what is your option about the cases you were not able to detect vibrio before the symptoms appear (Figure 1 - samples 3,7,9).
I would really recommend better explore the discussion of your data.
Moreover, you have used triplicates of your qpcr analyses but no statistical data is shown in your figures or mentioned in your text, you should include it.

Experimental design

No comments

Validity of the findings

Statistical data should be included in the figures.

Additional comments

Minor comments
Line 55. Add italic in Vibrionaceae word.
Line 79. I was wondering why did you used Zobell medium instead of TCBS that is selective for vibrio samples, or even TSA or MA?
It seems you use the analysis of BRD in 50 clams as sample size, but indeed the size of your sample analysis were bigger. Add this comment in the line 93.
It seems you have used 12 samples from each tree different conditions to totalize Thirty six clams sampled at each different time point. Add this information in line 105. This information will be very useful to correlate the 12 samples presented in your figures.
Line 216. You should add the Standard deviation value.
Line 297. Delete “()”.

---

## Round 0.2 · accepted · Accept

Based on the revised version of the your manuscript it has been accepted for publication.